# Mining Salt Tolerance SNP Loci and Prediction of Candidate Genes in the Rice Bud Stage by Genome-Wide Association Analysis

**DOI:** 10.3390/plants12112163

**Published:** 2023-05-30

**Authors:** Rui Wang, Zhenzhen Zhou, Mengyuan Xiong, Mingyu Du, Xingxing Lin, Cuiping Liu, Mingwei Lu, Zhengbo Liu, Yinping Chang, Erbao Liu

**Affiliations:** College of Agronomy, Anhui Agricultural University, Hefei 230036, China; 20113933@stu.ahau.edu.cn (R.W.); 20113943@stu.ahau.edu.cn (Z.Z.); 17766105110@stu.ahau.edu.cn (M.X.); 2541758057@stu.ahau.edu (M.D.); lxx20113884@stu.ahau.edu.cn (X.L.); lcp20113882@stu.ahau.edu.cn (C.L.); lmw20020627@stu.ahau.edu.cn (M.L.); lzb514609542@stu.ahau.edu.cn (Z.L.); changyinping@stu.ahau.edu.cn (Y.C.)

**Keywords:** rice, germination stage, salt tolerance, SNP loci, genome-wide association analysis

## Abstract

Mining salt tolerance genes is significant for breeding high-quality salt-tolerant rice varieties in order to improve the utilization of saline–alkaline land. In this study, 173 rice accessions were measured for their germination potential (GP), germination rate (GR), seedling length (SL), root length (RL), germination potential relative to salt damage rate (GPR), germination rate relative to salt damage rate (GRR), seedling length relative to salt damage rate (SLR), relative salt damage rate at the germination stage (RSD) and comprehensive relative salt damage rate in the early seedling stage (CRS) under normal and salt stress conditions. Genome-wide association analysis was performed with 1,322,884 high-quality SNPs obtained by resequencing. Eight quantitative trait loci (QTLs) related to salt tolerance traits at the germination stage were detected in 2020 and 2021. They were related to the GPR (*qGPR2*) and SLR (*qSLR9*), which were newly discovered in this study. Three genes were predicted as salt tolerance candidate genes: *LOC_Os02g40664, LOC_Os02g40810*, and *LOC_Os09g28310*. At present, marker-assisted selection (MAS) and gene-edited breeding are becoming more widespread. Our discovery of candidate genes provides a reference for research in this field. The elite alleles identified in this study may provide a molecular basis for cultivating salt-tolerant rice varieties.

## 1. Introduction

Rice (*Oryza sativa* L.) is an important staple food that feeds approximately half of the world’s population. It is a non-halophytic, salt-intolerant gramineous plant that evolved from freshwater marsh plants [1]. The term non-halophytic refers to salt-sensitive plants, due to the fact that salt-sensitive plants can experience sublethal or lethal effects with only slight increases in salinity [1,2]. Therefore, rice, as a non-halophytic plant, needs improvements to its salt tolerance. Therefore, salt stress is one of the most severe abiotic stress factors that limits the growth and development of rice, and rice plants are affected to varying degrees during different growth stages, mainly during the germination, seedling, and reproductive growth stages [3]. In recent years, industrialization and urbanization have accelerated, and the ecological environment has become increasingly harsh. Long-existing saline–alkaline lands have not yet been effectively improved and developed, and secondary soil salinization has further affected the growth conditions of crops, leading to a continuous reduction in grain yields [4]. Therefore, one way to increase rice yields is to fully utilize saline–alkaline land to produce rice. The studies of salt tolerance during the germination stage of rice and the mining of salt tolerance genes are of great significance for screening salt-tolerant germplasms, cultivating salt-tolerant rice varieties, and promoting the development and utilization of saline–alkaline land [5,6].

A large number of related studies have been conducted due to the importance of salt tolerance in rice. According to previous research, salt tolerance is a polygenic trait controlled by quantitative trait loci (QTLs) [7]. To date, approximately 439 QTLs related to salt tolerance have been identified [8,9,10,11,12], and at least 79 genes associated with the salt tolerance of rice have cloned [13,14,15,16,17]. All of these genes can be used as germplasm resources for cultivating salt-tolerant rice varieties. Marker-assisted selection (MAS) and gene-edited breeding are effective breeding methods. The rice accessions containing salt tolerance genes can be used as the elite parents for pyramiding breeding by MAS. The elite alleles of these genes can also provide the precision information for gene-edited breeding. Among these genes, 13 salt tolerance genes have been identified as having biological functions, which can be divided into the following three categories.

The first category of genes, including *OsCOIN*, *OcCPK21*, *OsHBP1b*, *OsGASR1*, *OsYUC1*, and *LOC_Os10g04860*, regulate salt tolerance in rice by regulating hormone metabolism. *OsCOIN* enhances the resistance of rice to salt via the overexpression of zinc finger proteins and increased proline contents. The expression of this gene is strongly induced by low temperature, abscisic acid (ABA), salt, and drought. The seeds from the *OsCOIN* transgenic rice germinated faster (35%) and grew taller than that from the wild-type rice [18]. *OsCPK21* promotes responses to ABA and salt stress through phosphorylation and participates in the positive regulation of ABA and salt stress responses. Overexpression of *OsCPK21* conferred increased sensitivity to exogenous ABA, enhanced tolerance to salt stress, and increased expression of ABA and stress-inducible genes, suggesting that *OsCPK21* is involved in ABA and salt stress signaling. Asano et al. [19] identified the positive mediation of signal pathways in response to ABA and salt stress based on the construction of a CDPK overexpression vector and the transformation of rice with this vector. Overexpression of *OsCPK21* confers increased sensitivity to exogenous ABA and enhances tolerance to salt stress. It could contribute to generating rice with improved tolerance to salt stress. In the process of studying the response of rice to abiotic stress, Lee et al. observed the wilting ratio, growth of new leaves, and changes in fresh weights, which showed that the ectopic expression of *OsGASR1* can enhance plant salt tolerance. OsGASR1 was linked with oxidative stress tolerance in *E.coli* cells that over-expressed OsGASR1 proteins. Therefore, OsGASR1 can act as a ROS-scavenging protein, consequently increasing salt stress tolerance in overexpression lines [20]. Cui et al. [21] conducted a correlation analysis of salt tolerance-related traits in 478 rice accessions in seed germination stages and found that *LOC_Os01g45760* and *LOC_Os10g04860* were involved in auxin biosynthesis, according to the enriched GO terms and KEGG pathways.

The second category of genes, including *SNAC2*, *OsZB8*, and *OsNAC5*, reinforce salt tolerance in rice by controlling the expression levels of transcription factors. *SNAC2* regulates salt tolerance in rice by encoding plant-specific *NAC* transcription factors, and the overexpression of *SNAC2* can significantly enhance salt tolerance by significantly increasing sensitivity to ABA [22]. The bZIP factor OsZB8, which is a component of the ABRE-DNA protein complex, plays an important role in the transcriptional regulation of rice vegetative tissues. The formation of the complex is upregulated in rice treated with NaCl, suggesting the regulation of *OsZB8* at both transcriptional and post-translational steps. There is a positive correlation between the expression of this gene and salt resistance. The higher *OsZB8* transcription is, the stronger the salt resistance [23]. Takasaki et al. [24] found that the expression of OsNAC5, a transcription factor specific to the plant NAC family, was induced in response to salt exposure. The salt tolerance of plants overexpressing *OsNAC5* increases significantly, indicating that the stress response protein OsNAC5, a transcriptional activator, improves plant resistance to salt stress by binding to the promoter and upregulating the expression of stress-tolerant genes, such as *OsLEA3* (a “late embryogenesis abundant” gene) [24].

The third category of genes, including *OsMAPK33*, *OsCLC1*, *OsHKT2;1*, *SKC1*, and *OsHKT1;5* modulate ion transport to affect the sensitivity of rice to salt stress. Lee et al. [25] evaluated the salt tolerance of *OsMAPK33* transgenic lines by measuring the water loss and osmotic potential of rice plants. The results of *OsMAPK33*-overexpressing lines showed more damage in growth and higher sodium uptake into the cell, resulting in a lower K^+^/Na^+^ ratio inside cells than wild-type or *OsMAPK33*-suppressed lines in the presence of salt stress [25]. The *OsCLC1* gene expressed in leaves and roots encodes a voltage-dependent Cl^−^ channel protein that can be instantaneously induced under salt stress. The co-expressed Na^+^/H^+^ reverse transporter *OsNHX1* and H^+^-ATPase subunit OsVHA-B jointly regulate the balance of anions and cations under salt stress and play an important role in maintaining osmotic pressure under high-salt conditions [26]. The *OsHKT2;1* gene, encoding a sodium ion absorption transporter protein, is the direct target gene of *OsPRR73*, as identified based on transcriptome analysis combined with biochemical evidence. The *OsPRR73* protein, which binds to the promoter of *OsHKT2*, inhibits the expression of *OsHKT2;1* in the temporal dimension and reduces the absorption of sodium ions within a specific time window to prevent the excessive accumulation of sodium ions. The *OsHKT2;1* gene, which is located downstream of *OsPRR73,* can regulate salt tolerance in rice by regulating sodium ion homeostasis and ROS levels. Salt-induced *OsPRR73* expression conferred salt tolerance by recruiting HDAC10, the nuclear interactor of OsPRR73 and co-repressor of *OsHKT2;1* [27]. The *SKC1* gene encodes a sodium ion transporter of the HKT family. The Na^+^ transporters of SKC1 play an important role in controlling K^+^ and Na^+^ transportation in rice plants under salt treatment by affecting K^+^ accumulation, which plays an important role in maintaining the K^+^/Na^+^ ratio and the tolerance of rice to salt [28]. The *OsHKT2;1* gene encodes a Na^+^-selective transporter, which is localized in cells adjacent to the xylem in roots. When rice is under salt stress, excessive Na^+^ is returned to the roots through the xylem, thereby reducing the harm caused by Na^+^ and enhancing the salt tolerance of rice [29].

Cloned salt tolerance-related genes provide a good functional basis for creating and cultivating new germplasm resources for salinity tolerance. However, the mining and cloning of relevant genes conducted to date have mainly been focused on the seedling stage and reproductive growth stage. In contrast, there are relatively few reports on the mining of salt tolerance genes in rice during the germination stage, and the molecular regulatory mechanism needs to be further analyzed. Therefore, the main goal of this study is to mine salt tolerance genes in rice and cultivate elite salt-tolerant rice germplasm resources. A total of 173 rice accessions from 5 countries were used in this experiment. The screening of salt tolerance genes and identification of elite alleles was conducted in the rice germination stage using SNP molecular markers. The variation in elite salt-tolerant alleles discovered in the germination stage provides molecular data and a theoretical basis for breeding high-quality and salt-tolerant rice varieties.

## 2. Results

### 2.1. Abundant Phenotypic Variation in Salt Tolerance-Related Traits in 173 Accessions

The frequency distribution histograms of salt tolerance-related traits during the rice germination stage in 2020 and 2021 are shown in the following figure (Figure 1). The distributions of germination potential (GP), germination potential relative to salt damage rate (GPR), seedling length (SL), and seedling length relative to salt damage rate (SLR) are all close to normal, and there are few outliers. Although the distribution of root length (RL) is also close to normal, it includes many extreme values.

In this study, wide variations in salt tolerance-related phenotypic traits were observed in the 173 rice varieties over the period from 2020 to 2021, with broad-sense heritability, ranging from 46.26% to 89.05% (Table 1). Broad-sense heritability was higher for traits such as GP, GPR, SL, SLR, RL, and root length relative to salt damage rate (RLR). Among these traits, germination rate relative to salt damage rate (GRR) showed the highest coefficients of variation, of 117.4% and 124.3% in 2020 and 2021, respectively, while germination rate (GR) showed the lowest coefficients of variation, of 15.27% and 10.26%. The coefficients of variation for other salt tolerance-related traits ranged from 29.32% to 81.19%.

Ten different rice varieties were selected to demonstrate the rich variation in salt tolerance during the germination stage (Figure 2) under 140 mmol/L NaCl salt stress treatment for ten days. The results showed that the 173 rice germplasm resources presented rich variation in salt tolerance-related traits at the germination stage, and this population was deemed suitable for mining elite allelic variations in salt tolerance-related traits at the germination stage.

An analysis of the correlations among 10 salt tolerance traits during the 2-year seedling stage (Table 2) showed that GPR and GRR were negatively correlated with GP and GR, while GPR, GRR, and relative salt damage rate at the germination stage (RSD) were positively correlated. SLR, RLR, and comprehensive relative salt damage rate in the early seedling stage (CRS) were positively correlated, while SLR, RLR, CRS, SL, and RL were negatively correlated. Among these traits, GPR, SLR, and CRS showed significant or extremely significant correlations with other traits in the correlation analysis, and these results can comprehensively reflect the growth stages of rice under salt stress at the germination stage and the affected stage. In previous studies, many different indicators of salt tolerance were selected. Among these indicators, relative traits between the control group and the salt stress group are reported as demonstrating high correlation with salt tolerance [30,31]. The CRS and RSD are comprehensive evaluation indicators of salt tolerance, which play important roles in the phenotypic investigation of salt tolerance. Qi et al. (2006) [32] showed that RL, SL, and RSD were significantly correlated. Therefore, based on the previous studies and the results of the correlations in this study, we can use GPR, SLR, CRS, and RSD as important indicators to analyze the genome-wide associations of salt tolerance genes.

### 2.2. Identification of 2 Subgroups of the 173 Accessions

The results of this experiment were based on the population structure analysis of 173 rice varieties, constructed using SNP markers. As shown in Figure 3b, the logarithmic likelihood value increased with an increase in the number of subgroups, indicating that the appropriate number of subgroups could not be determined. Therefore, the appropriate number of subgroups was determined based on the rate of change in the logarithmic likelihood value (∆K). As shown in Figure 3c, when the number of subgroups was equal to 2, the ∆K value was the largest, indicating that the natural population of 173 rice varieties can be divided into 2 subgroups. In Figure 3, these 2 subgroups were named Pop1 and Pop2, corresponding to 108 and 65 varieties, respectively. In order to further verify the above conclusion, principal component analysis and Nei’s genetic distance clustering analysis were conducted. As shown in Figure 3d,e, the 173 rice varieties were divided into 2 groups, consistent with the results of the structure analysis. Thus, the high reliability of dividing the core germplasm population into two subgroups was further verified.

### 2.3. Linkage Disequilibrium

In this study, 1,322,884 SNPs obtained through filtering with PLINK software were distributed across all 12 chromosomes (Figure 4a). Linkage disequilibrium (LD) analysis was performed using these 1,322,884 SNP loci, and the results are shown in Figure 4b. The corresponding decay distance was calculated as 90.11 kb, when the R^2^ value declined to the maximum value of 1/2, which was 0.4323.

### 2.4. Twelve SNP Loci Related to Salt Tolerance Detected in Two Study Years

Under the condition of setting the significance threshold at *p* < 3.78 × 10^−4^, a genome-wide association analysis was performed using a general linear model (GLM) and mixed linear model (MLM) to associate these SNP markers with salt tolerance-related traits, including GP, GPR, SL, SLR, RL, and RSD. The mixed linear model with kinship correction was chosen based on its strength and accuracy in the model calculations. Based on the results of GPR, SLR, and RSD, in total, 12 significantly associated SNP loci (Table 3) were detected in the two years, and the contribution rates (R^2^) of the SNP loci ranged from 10.1% to 12.8% (Table 3). No significantly associated SNP loci were detected for CRS. The significantly associated SNP loci with other traits were showed as Appendix A.

In the 2 study years, a total of 12 (6 for GPR, 2 for SLR, and 4 for RSD) significant QTLs were significantly associated with GPR and SLR. These QTLs were located on chromosomes 1, 2, 3, 6, 7, 9, and 12, explaining from 10.1% to 12.8% of the phenotypic variation. For GPR, six QTLs were detected on chromosomes 1, 2, 6 and 7. Among them, *qGPR1-2* (Chr1_38,309,966) contributed the largest phenotypic variation, of 12.8%. Additionally, the *qGPR2-1* SNP locus was identified in both 2020 and 2021, explaining 10.4% and 11.8% of the phenotypic variation, respectively. For SLR, *qSLR2* (Chr2_9,878,862) and *qSLR9* (Chr2_17,238,271) were identified. The *qSLR2* (Chr2_9,878,862) locus explained 10.10% of the phenotypic variation, and the *qSLR9* SNP locus was also identified in both years, explaining up to 11.6% of the phenotypic variation. For RSD, total four QTLs were detected. The *qRSD3* (Chr3 5,919,379) locus showed the highest contribution (12.6%) to the phenotypic variation. Additionally, *qRSD2*, detected on Chr2, showed the lowest phenotypic variation. All QTLs detected for RSD were only detected in one year. Therefore, *qGPR2-1* and *qSLR9*, which were both detected in both years, are predicted as the SNP loci related to salt tolerance during the germination stage that were identified for the first time in this study.

### 2.5. Prediction of Salt-Tolerant Candidate Genes at the Germination Stage

Within the 90 kb upstream and downstream regions of *qGPR2-1* and *qSLR9*, candidate genes were identified using the Rice Genome Annotation Project (Rice Genome Annotation Project Rice Genome Browser—Release 7) database (Appendix A). In total, 53 candidate genes were identified in the two regions, including 3 hypothetical proteins, 5 retrotransposon proteins, 2 transposon proteins, 16 uncharacterized expressed proteins, and 27 functionally annotated genes (Appendix A).

The MSUIDs of the 53 genes in the two regions were converted via the National Rice Data Center resource to list the relevant Gene IDs, and the SNP information of the 173 rice materials was employed to find the base mutation regions of these genes using Git (Git git-scm.com). The exon regions of these genes were checked in the National Center for Biotechnology Information (nih.gov, accessed on 20 September 2021) database in order to examine whether nonsynonymous mutations occurred. After sorting out the information, nonsynonymous mutations were found in 15 gene exons (Table 4).

These 15 genes comprised 10 cloned and 5 uncloned genes. Among the cloned genes, four confer resistance, including *LOC_Os02g40770*—which is involved in protein methylation, influences chromosome structure, and regulates gene expression [34]—and *LOC_Os02g40784*, which improves plant immunity and provides durable resistance genes. It can also enable the development of resistant rice cultivars [35]. The gene overexpression factor *OsbZIP72*, of *LOC_Os09g28310*, activates the rice high-affinity potassium transporter *OsHKT1*, and thereby participates in transcriptional gene regulatory pathways of ABA signal transduction and mediates salt tolerance in rice [36]. The functional annotation of the uncloned genes indicated that *LOC_Os02g40664* and *LOC_Os02g40810* are zinc finger family proteins that are possibly linked to salt tolerance. Thus, *LOC_Os02g40664*, *LOC_Os02g40810*, and *LOC_Os09g28310* are predicted candidate genes for salt tolerance in rice at the germination stage.

### 2.6. Elite Haplotype Analysis

We selected three genes and further analyzed the haplotypes of these genes (Figure 5). For the *LOC_Os02g40664* gene (Figure 6) at the *qGPR2* locus, all germplasms can be divided into six haplotypes based on an SNP in its cDNA. The statistical analysis of the relative salt damage rates of four traits in 2019 and 2020 showed that there was no significant difference among the six haplotypes for GP and GR, whereas there were significant differences among the haplotypes for SL and RL. The average values of the six haplotypes for SLR ranged from 41.9 ± 17.3% to 66.7 ± 12.5%, and the average values of the six haplotypes for RLR ranged from 46.5 ± 34.3% to 93.8 ± 8.8%. Among the six haplotypes, HapC was the elite haplotype of *LOC_Os02g40664*, which was associated with the lowest average SLR and RLR values among the six haplotypes.

We analyzed the SNPs of cDNA exons at the *qSLR9* locus and found that *LOC_Os02g40810* included five nonsynonymous SNPs (Figure 7). The materials were divided into five haplotypes, based on the SNP analysis of cDNA in all germplasms. The statistical analysis of relative salt damage rates was conducted for five traits in 2019 and 2020, and revealed no significant differences among the five haplotypes for SL and GR. However, significant differences were observed among the haplotypes for GP and RL. The average values of the five haplotypes for GPR ranged from 29.0 ± 33.8% to 57.6 ± 34.8%, whereas for RLR, the average values of the five haplotypes ranged from 42.3 ± 28.8% to 85.7 ± 18.0%. For GPR, HapA, HapB, HapD, and HapE were the elite haplotypes. For RLR, HapA was the elite haplotype. Therefore, HapA is an elite haplotype for both GPR and RLR, indicating that HapA, of *LOC_Os02g40810*, is important for salt tolerance.

For the gene located at the *qSLR9* locus, *LOC_Os09g28310* (*OsbZIP72*), which has been reported to be related to salt tolerance in rice [36], four haplotypes were identified based on the SNP analysis of cDNA in all germplasms (Figure 8). The statistical analysis of relative salt damage rates was conducted on four traits in 2019 and 2020, revealing no significant differences among the haplotypes for GR. However, significant differences were observed among the haplotypes for SL, GP, and RL. The average values of the four haplotypes for GPR ranged from 26.3 ± 25.3% to 50.3 ± 32.4%; for RLR, the average values of the four haplotypes ranged from 70.2 ± 34.8% to 90.3 ± 15.6%; for SLR, the average values of the haplotypes ranged from 56.9 ± 17.0% to 67.4 ± 16.1%. For GPR, HapA showed the lowest average GPR among all haplotypes. For SLR, HapA, HapB, and HapD were the elite haplotypes. HapD was an elite haplotype for RLR. Therefore, HapA and HapD, of *LOC_Os09g28310* (*OsbZIP72*), may play key roles in improving salt tolerance at the bud stage.

## 3. Discussion

In order to fully demonstrate the variation in the population of 173 accessions under salt stress during the germination stage, 10 rice varieties were selected based on different salt concentration treatments, including 5 indica rice varieties and 5 japonica rice varieties. Under treatments involving four different salt concentrations (50 mmol/L, 80 mmol/L, 110 mmol/L, and 140 mmol/L), the growth status and salt tolerance-related traits of each variety were measured at 2 d, 4 d, and 10 d of treatment. The results showed that the phenotypic differences among the varieties were largest and the coefficient of variation was highest at a salt concentration of 140 mmol/L among the tested treatments. Therefore, this study selected 140 mmol/L as the most suitable concentration for salt stress treatment during the germination stage. At this concentration, the coefficient of variation of salt tolerance-related phenotypes in the core germplasm population reached a maximum of 124.3%. In the case of a relatively small population size, the appropriate salt concentration treatment provided a good foundation for the comprehensive exploration of salt tolerance-related genes during the germination stage in this study.

The linkage disequilibrium analysis of 173 rice core germplasm resources showed that the population decay distance was 90 cM. This LD decay rate is faster than those recorded in other studies, possibly due to artificial hybridization and recombination of the core germplasm accessions. However, the rice attenuation distance is greater than that of cross-pollinated crops because rice is a self-pollinating plant. Compared to traditional linkage mapping methods, association analysis uses different materials, and the contribution rate of allelic variation to a phenotype is determined based on different varieties rather than on parents [37]. If the group structure is not considered in the association location, it can easily lead to the occurrence of type I errors [38]. Therefore, it was necessary to analyze the accession structure of the 173 rice core collections. We determined that the accessions can be divided into two subgroups through structure analysis. Accurate population structure analysis laid a foundation for accurate association analysis.

In this two-year study, *qGPR2-1* and *qSLR9* were the QTLs detected in both years. Mohammadi et al. [39] located four grain yield QTLs on chromosomes 2, 4, 6, and 8 under salt stress during the reproductive stage using QTL mapping in an F_2_ population of Sadri/FL478 (IR96469) hybrids, which explained 31.6% of the observed phenotypic variations. Among these QTLs, *qGY4.1*, which affects the grain yield, was located between RM551 and RM518 (2.03 Mb) on chromosome 4, which was adjacent to the interval of *qSL4* (4.59 Mb), located in this study (Appendix A) in the study year 2020 [39]. Lin et al. identified four QTLs located on chromosomes 1, 3, 5, and 11 using recombinant inbred lines (RILs) derived from the local indica rice variety Zhaxima in Yunnan Province and the high-quality japonica rice variety Nanjing 46 in Jiangsu Province, and germination stage salt tolerance was used as the considered phenotype value. Among these QTLs, the interval of *qSST11* (2.84–4.07 Mb) partially overlapped with the interval of *qRL11.1* (3.50 Mb) (Appendix A), which was located in this study in 2020 [40]. Thomson et al. [16] located *qSES4* between RM3843 and RM127 (31.49–34.52 Mb) on chromosome 4 using 100 SSR markers in 140 IR29/Pokkali RILs, which partially overlapped with the interval of *qSL4* (33.31 Mb) (Appendix A), located in this study in 2021. Wang et al. [12] located *qSH12.2* between RM7376 and RM6953 (23.44–26.12 Mb) on chromosome 12 under three different salt concentration treatments (0.0%, 0.5%, and 0.7% NaCl) using an F_2:3_ hybrid population, which partially overlapped with the interval of *qRSD12* (23.87–25.90 Mb) (Table 3), and which was located in this study in 2021. Qiu et al. [41] identified a salt tolerance allele (*qSST1.1*) on chromosome 1 in two groups of backcross introgression line (IL) populations between RM3143 and RM443 (26.82–28.33 Mb), which was located in the interval of *qGP1.1*, identified in this study in both 2020 and 2021 (25.27–42.64 Mb) (Appendix A). The *qGPR2-1* locus, identified in this study on chromosome 2, and *qSLR9*, on chromosome 9, did not contain any cloned genes, and thus represent new loci identified in this study.

In *qGPR2-1*, we identified *LOC_Os02g40664* and *LOC_Os02g40810* as promising candidate genes, both of which were annotated as zinc finger family protein genes. Zinc finger proteins are crucial regulatory factors that respond to nonbiological stress in plants. The first C2H2-type zinc finger protein gene identified in plants was EPF1 from Arabidopsis, which encodes a protein with two typical ethylene zinc finger motifs [42]. C2H2-type zinc finger proteins are involved in enhancing salt tolerance by influencing salt-regulated genes, maintaining ion homeostasis, increasing the concentrations of osmotic adjustment substances, and improving the ability to clear ROS. Many C2H2-type zinc finger proteins enhance plant salt tolerance through the ABA-mediated signaling pathway. Within *qSLR9*, we predicted *LOC_Os09g28310* to be a candidate gene. The transcription factor *OsbZIP72*, encoded by this gene, has been expressed and integrated into a transgenic rice plant genome. Wang et al. [12] found that *OsbZIP72* enhances the transcriptional regulation pathway of ABA signal transduction by activating the high-affinity potassium transport protein *OsHKT1;1* in rice under salt, ABA, and drought stresses [36]. The overexpression of *OsbZIP72* in plants resulted in increased tolerances to drought and salt stress. Therefore, these three candidate genes are highly likely to be involved in regulating salt tolerance during the germination stage of rice.

The distribution of candidate gene haplotypes in two identified subpopulations showed that *LOC_Os02g40664* and *LOC_Os02g40810* exhibit significant indica/japonica differentiation in several major haplotypes (Appendix A). The elite haplotypes of *LOC_Os02g40664* and *LOC_Os02g40810* were mainly distributed in indica rice. The elite haplotypes of *LOC_Os09g28310* were mainly distributed in japonica rice (Appendix A). All of the rice carrier varieties harboring elite haplotypes for salt tolerance in the bud stage exhibited a significantly improved salt tolerance in the bud stage. By continuously aggregating allelic variations related to salt tolerance, the salt tolerance of rice varieties can be greatly improved, thereby achieving the goal of planting and increasing the yield of rice in saline–alkaline land. Rice varieties containing the elite allelic variations mentioned above can serve as excellent parents in the cultivation of salt-tolerant rice varieties, providing good genetic resources for the successful cultivation of salt-tolerant rice varieties.

Mining salt tolerance SNP loci and the prediction of candidate genes by GWAS in this study were only focused on the rice bud stage. In fact, the salt tolerance at the seedling, reproductive, and other growth stages are also extremely important for increasing rice yields in saline–alkaline land [43]. Therefore, based on this study on the bud stage, our future studies will explore salt tolerance at different growth stages in order to fully understand the genetic mechanisms underlying salt tolerance in rice. Additionally, analyzing the interaction between QTL and the environment is also important for providing a scientific basis for the rice genetic mechanism of salt and alkaline tolerance and molecular marker-assisted breeding [44]. The contribution of environmental factors to salt tolerance-related traits was not considered in this study. Therefore, revealing the genetic mechanism of panicle number per plant, seed setting rate, thousand-grain weight, and panicle weight per plant in rice under salt and alkaline stress are necessary, and it would be interesting to investigate the interaction between genetic and environmental factors in future studies.

## 4. Materials and Methods

### 4.1. Plant Materials

We selected 173 different rice varieties with abundant variation in salt tolerance from multiple regions and countries in Asia as the materials for this study (Appendix A). Among these resources, 134 rice germplasm resources came from 13 provinces or cities in China, including Heilongjiang, Tianjin, Henan, Yunnan, Hunan, Hubei, Sichuan, Guangdong, Hainan, Fujian, Jiangsu, Anhui, and Shanghai, and the other 39 rice germplasm resources originated from Japan, Indonesia, the Philippines, and Vietnam (detailed information in Appendix A).

### 4.2. Field Planting and Management

All 173 accessions were planted at the Jiangpu Experimental Station of Nanjing Agricultural University in Nanjing, China, over two years (2019–2020). The germplasms were sown on 17 May and transplanted to the field on 20 June. Each variety was planted in one plot, with 3 rows and 8 plants per row. The distance between each plant was 16.7 cm (5 inches), and the distance between each row was 26.7 cm (8 inches). The plants were transplanted individually and managed using conventional field management practices. After maturity, 10 panicles of each variety were mixed and dried, and the seeds were then placed in a 50 °C drying oven for 3 d to break dormancy before being used for salt tolerance phenotype determination during the germination period.

### 4.3. Identification of Salt Tolerance

A salt concentration sensitivity test was conducted, involving different salt concentrations, and showed that the variation in salt tolerance of the population was the largest with 140 nmol/L NaCl treatments. In total, 180 normal rice seeds with the same size, plumpness, and color were selected and divided into a CK control group (distilled water as control) and a salt stress control group (140 mmol/L salt concentration). Each group included 3 replicates, with 30 rice seeds per replicate. The seeds were surface sterilized with 0.5% sodium hypochlorite solution for one hour and rinsed with water three times. Sterilized seeds were placed on germination paper in a Petri dish, with 30 seeds in each Petri dish, and 10 mL of distilled water or salt solution was added to each Petri dish in the control and treatment groups, respectively. Each dish was refilled with fresh nutrient solution every two days, while they were incubated in darkness with 70% humidity. The germination number was recorded on the fourth day after the seeds were placed in the incubator. GP was calculated by taking half of the length of rice seeds at germination as the germination standard; the germination number was recorded on the tenth day. GR was calculated based on the number of germinated seeds. In addition, five plants were randomly selected from each Petri dish in order to measure their SL and RL, and calculate GPR, GRR, SLR, RLR, RSD, and CRS. The above traits were calculated as follows:GP (%) = (number of germinated grains on the 4th day/total number of grains) × 100%GR (%) = (number of germinated grains on the 10th day/total number of grains) × 100%GPR (%) = (control germination potential-treatment germination potential)/control germination potential × 100%GRR (%) = (control germination rate-treatment germination rate)/control germination rate × 100%SLR (%) = (control seedling length-treated seedling length)/control seedling length × 100%RLR (%) = (control root length-treated root length)/control root length × 100%RSD (%) = (GPR + GRR)/2CRS (%) = (SLR + RLR)/2

### 4.4. Genotype Identification

Prior to tillering, 173 leaf samples were collected, and total DNA was extracted using the standard CTAB method. Five micrograms of whole genomic DNA was extracted from each variety and used to construct paired-end sequencing libraries with an insert size of approximately 350 bp. Paired-end 150 bp reads were generated using the Illumina HiSeq X10 platform. The raw sequences were further processed in order to remove adapter contamination, reads with a N ratio greater than 10%, reads with all A base, and low-quality reads (alkali base with mass value Q ≤ 20 accounts for more than 50% of the entire read), resulting in 0.532 TB of genome sequence data with an average coverage depth of 5.48X per genome. Library construction and sequencing were performed by Meiyin Health Technology (Beijing) Co., Ltd., Beijing, China. For the sequencing data, using Nipponbare as the reference genome, SNP detection was performed using GATK 4.0, and PLINK software was used to filter out population SNPs with a minimum allele frequency (MAF) of greater than 5% and missing rate of less than 20% for subsequent analysis. Thereafter, the gVCF file was generated after SNP Calling. Multiple sample gVCF files were merged to generate a VCF file, the data in which were used as the genotype data in this study.

### 4.5. Phenotypic Data Analysis

All phenotypic data were analyzed using Office365 (Enterprise Edition), and the obtained data were analyzed by SPSS 22.0 software. GraphPad Prism 8.0 software was used for statistical analysis of the data and correlation analysis among traits while calculating the standard deviation, maximum value, minimum value, coefficient of variation, and generalized heritability.
$$h_{B}^{2}=\frac{V_{G}}{V_{P}}\times 100=\frac{MS_{v}-MS_{e}}{MS_{v}+\left(r-1\right)MS_{e}}\times 100\left(\%\right)$$

*MS_v_* is genetic variance; *MS_e_* is error variance; *r* is the number of duplicates.

### 4.6. Structure Analysis

According to linkage disequilibrium, PLINK software was used to filter SNPs, and 38,482 SNP loci without linkage (window of 50 kb, step size of 10, correlation threshold of 0.2) were retained and converted into the corresponding STRUCTURE format [45]. The parameters used for the PLINK software were listed in Appendix A. The number of subgroups (K) was set to 1–10, and the number of random seeds was set to 1–3. The ‘define BURNIN’ was set as 5000 times, and ‘define NUMBERS’ was set as 50,000 times in the configuration file. We compressed and uploaded the results to the Structure Harvest website for online analysis (http://taylor0.biology.ucla.edu/structureHarvester, accessed on accessed on 15 November 2021). However, if the logarithmic likelihood increases with the number of subgroups, the change rate of logarithmic likelihood (ΔK) was proposed by Evanno et al. [46]. The parameters used for the Structure software were listed in Appendix A. The optimal number of subgroups was analyzed by website (http://taylor0.biology.ucla.edu/structureHarvester, accessed on accessed on 19 November 2021). Thereafter, visualize Structure results were conducted through R language.

The vcf2phylip.py script was used to convert the filtered SNP data, and the website ATGC: Montpellier Bioinformatics platform (atgc-montpellier.fr, accessed on 25 November 2021) was used to construct the neighbor-joining clustering map [47]. A high-quality NJ tree was prepared using the iTOL (https://itol.embl.de/, accessed on 30 November 2021) online tool [48].

### 4.7. Linkage Disequilibrium Analysis

In LD analysis, r^2^ equals 0 indicates that the population is in complete linkage equilibrium, and r^2^ equals 1 indicates that the population is in complete linkage disequilibrium. PLINK was used for LD analysis of genotype data, with missing genotypes set as 0 and minor allele frequency (--MAF) set as 0.05. The number of SNPs in the window (--LD-window) was 999,999, and the window size (--LD-window-Kb) was 1000. The minimum SNP (--LD-window-r^2^) in the window is 0, and the r^2^ value is output. The R package “qq-plot” was used to draw the LD heatmap [49,50,51].

### 4.8. Genome-Wide Association Analysis

Genome-wide association study (GWAS) analysis was performed on SNPs and phenotypes using the GLM and MLM in TASSEL software [52,53]. The Bonferroni correction method [54] was used to set the threshold for *p* values for GLM at 3.78 × 10^−a^ (i.e., 0.05/1,322,884). This method uses a *p*-value of less than 0.05 as the threshold to determine whether it is significant in the multiple significance tests. However, when comparing multiple sets of data simultaneously, simply using 0.05 as the threshold may not be appropriate. For significant associations, the Benjamini–Hochberg [55] correction method was used to calculate the false discovery rate (FDR), with a threshold of 1.0 × 10^−5^ for MLM. This method aims at solving the problem of multiple significance testing, which calls for controlling the expected proportion of falsely rejected hypotheses. It provides the potential for a gain in power over problems where there is control of the false discovery rate rather than that of the familywise error rate. SNPs within the same LD region were considered to represent a single QTL, where the lead SNP was that with the smallest *p* value. Manhattan plots were generated using R (CM plot).

### 4.9. Candidate Gene Prediction

Candidate genes were predicted by screening the germplasm resources for genes with nonsynonymous mutations among all genes in the identified linkage regions of the SNP loci. The MUS website (http://rice.plantbiology.msu.edu/index.shtml, accessed on 10 December 2021) was used for gene annotation, and previous research was considered to aid in candidate gene prediction.

### 4.10. Elite Haplotype Analysis

Candidate genes and all nonsynonymous SNPs in exons were selected for haplotype analysis. The haplotype distribution of candidate genes was analyzed according to geographical regions and subgroups. The average positive (negative) haplotype effect (AHE) within a gene locus was calculated as follows:AHE=∑hcnc

In the text, h_c_ represents the phenotype value of the c-th haplotype with positive (negative) effects, and n_c_ represents the number of haplotypes with positive (negative) effects within the base position. The rice accessions with the highest positive haplotype effects on all salt tolerance in rice during the germination stage were predicted to be the most promising parents for elite predecessors of salt tolerance at the germination stage.

## 5. Conclusions

In this study, we conducted SNP marker analysis of salt tolerance-related genes in 173 rice varieties using association analysis. Based on 2 years of GWAS results, 30 SNP loci associated with salt tolerance traits were identified, including *qGPR2-1* and *qSLR9*, located on chromosomes 2 and 9, respectively. These were identified as two salt tolerance QTLs at two seedling stages in this study. Three candidate genes (*LOC_Os02g40664*, *LOC_Os02g40810*, and *LOC_Os09g28310*) were predicted. Through candidate gene haplotype analysis, carrier materials containing two elite allele genes were selected. These research results provide a good molecular basis and germplasm resources for breeding salt-tolerant rice varieties.

## Figures and Tables

**Figure 1 plants-12-02163-f001:**
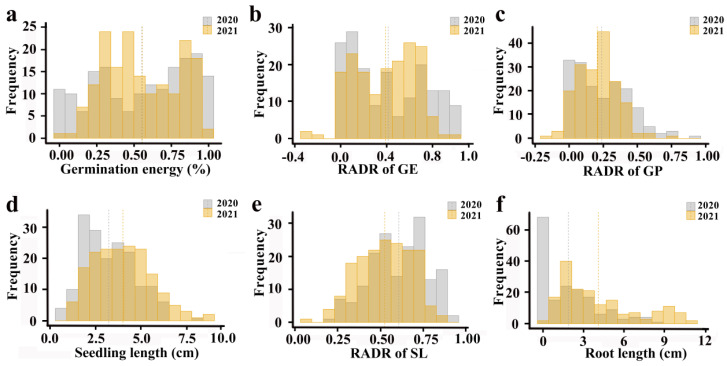
Distribution frequency histogram of salt tolerance-related traits of 173 rice accessions in two years. (**a**) Germination potential; (**b**) germination potential relative to salt damage rate; (**c**) seedling length; (**d**) seedling length relative to salt damage rate; (**e**) relative salt damage rate at the germination stage; (**f**) root length.

**Figure 2 plants-12-02163-f002:**
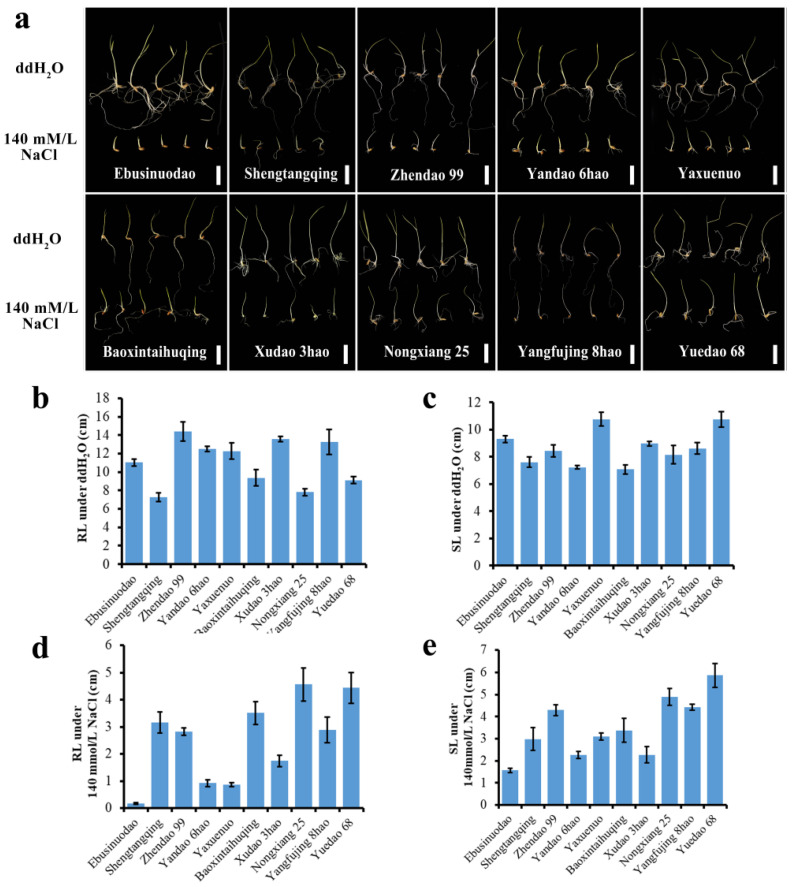
Phenotypic map of ten rice varieties with different salt tolerance levels selected after ten days of 140 mmol/L NaCl salt stress treatment. (**a**) Phenotype of root length and seedling length of ten rice varieties. Scale bar = 2 cm; (**b**) the variation in RL of ten varieties under ddH_2_O; (**c**) the variation in SL of ten varieties under ddH_2_O; (**d**) the variation in RL of ten varieties under 140 mmol/L NaCl; (**e**) the variation in SL of ten varieties under 140 mmol/L NaCl. RL: root length; SL: seedling length.

**Figure 3 plants-12-02163-f003:**
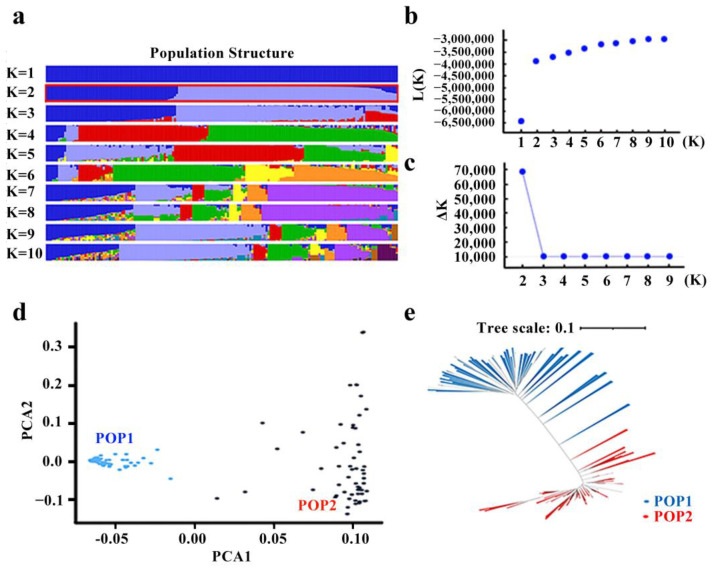
Population structure analysis of natural populations constructed based on 173 rice varieties. (**a**) Stacked histogram of group structure division; (**b**) changes in the log-likelihood function value based on the number of subpopulations; (**c**) changes in the number of subpopulations; (**d**) the results of principal component analysis; (**e**) neighbor-joining tree constructed according to Nei’s genetic distances of 173 rice accessions.

**Figure 4 plants-12-02163-f004:**
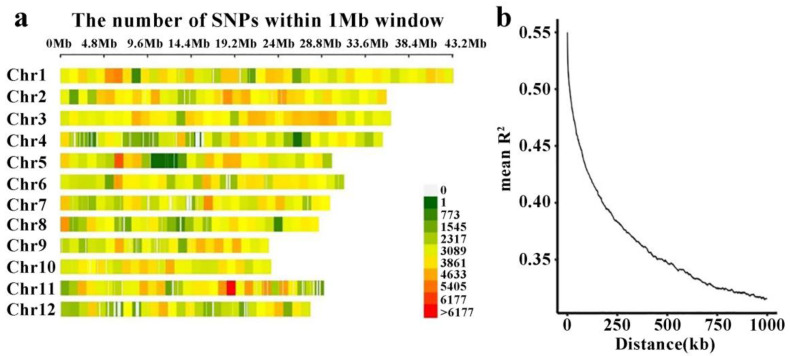
Analysis of 173 rice germplasms. (**a**) Chromosomal SNP distribution of 173 rice accessions; (**b**) linkage disequilibrium decay map.

**Figure 5 plants-12-02163-f005:**
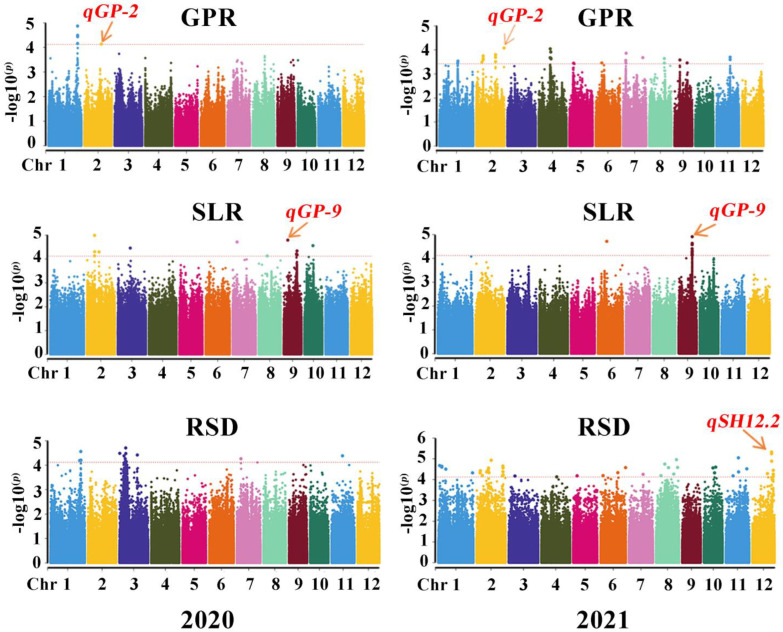
Manhattan plot of 173 rice germplasms obtained from salt tolerance germination experiments based on SNP markers (2020). Reported QTLs are indicated in the figure.

**Figure 6 plants-12-02163-f006:**
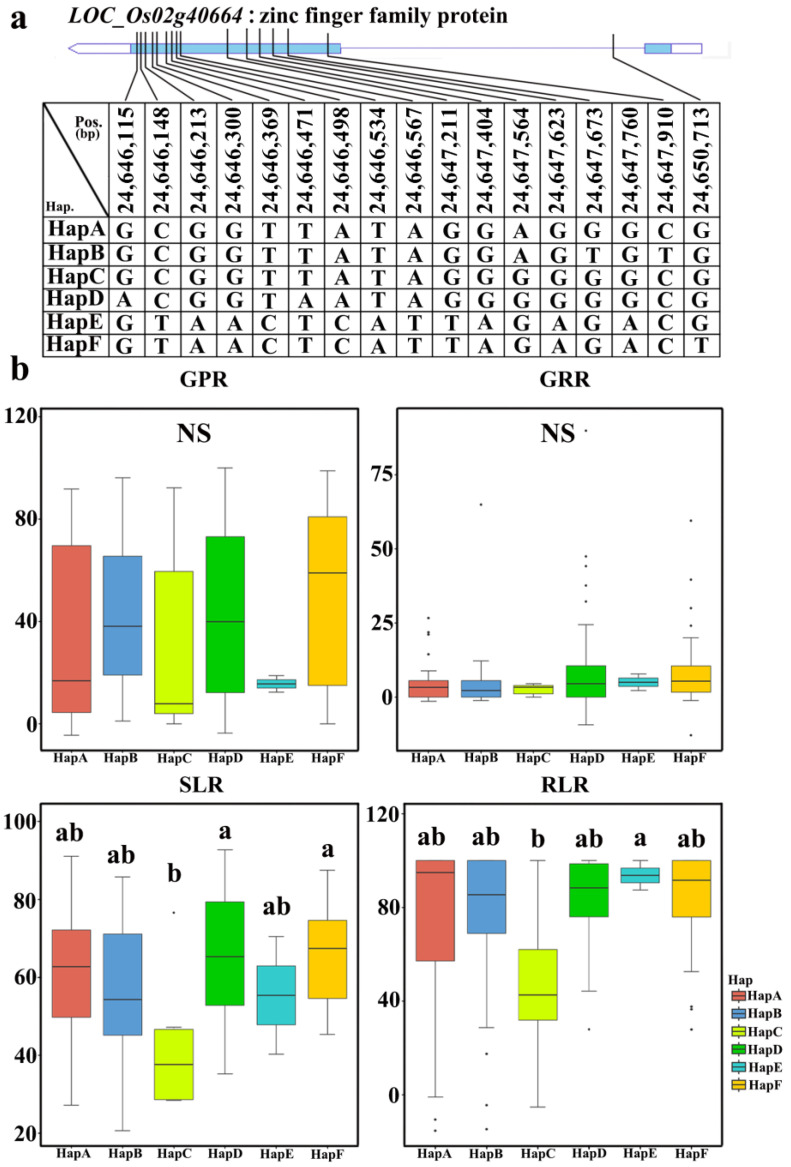
Haplotype analysis of a candidate gene: (**a**) Schematic representation of *LOC_Os02g40664* structure and single-nucleotide polymorphisms in *LOC_Os02g40664* cDNA among HapA, HapB, HapC, HapD, HapE, and HapF. Blue boxes indicate exons. (**b**) Box plots for GPR, GRR, SLR, and RLR in the six haplotypes of *LOC_Os02g40664* in all accessions in 2020. Central lines indicate the median value, and box edges represent the upper and lower quartiles. Multiple comparison tests were based on Duncan’s test at *p* < 0.05. Different lower case letters above bars indicate significant differences among haplotypes.

**Figure 7 plants-12-02163-f007:**
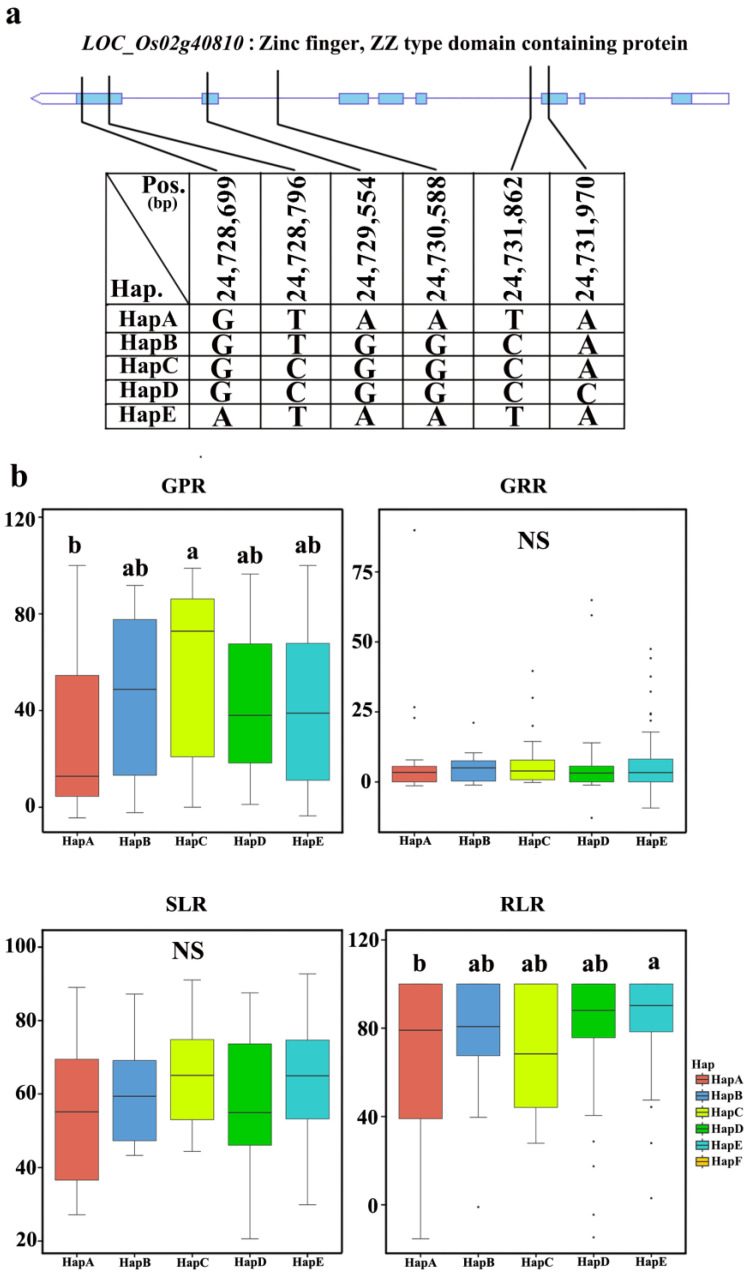
Haplotype analysis of a candidate gene: (**a**) Schematic representation of *LOC_Os02g40810* structure and single-nucleotide polymorphisms in *LOC_Os02g40810* cDNA among HapA, HapB, HapC, HapD, and HapE. Blue boxes indicate exons. (**b**) Box plots for GPR, GRR, SLR, and RLR in the five haplotypes of *LOC_Os02g40810* in all accessions. Central lines indicate the median value, and box edges represent the upper and lower quartiles. Multiple comparison tests were based on Duncan’s test at *p* < 0.05. Different lower case letters above bars indicate significant differences among haplotypes.

**Figure 8 plants-12-02163-f008:**
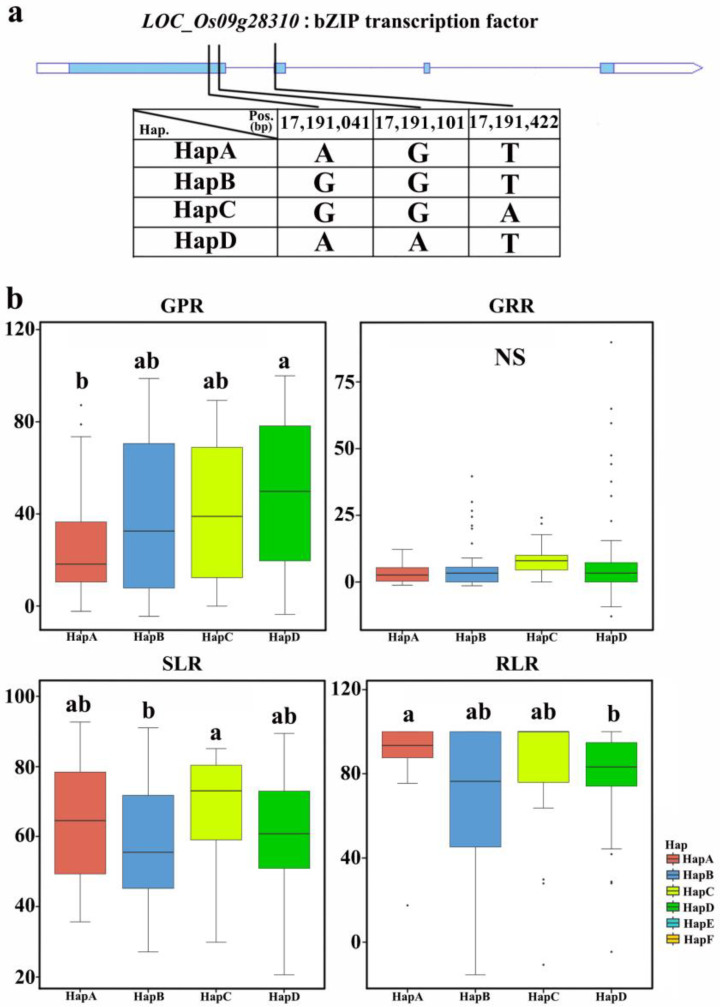
Haplotype analysis of a candidate gene: (**a**) Schematic representation of *LOC_Os09g28310* structure and single-nucleotide polymorphisms in *LOC_Os09g28310* cDNA among HapA, HapB, HapC, and HapD. Blue boxes indicate exons. (**b**) Box plots for GPR, GRR, SLR, and RLR in the four haplotypes of *LOC_Os09g28310* in all accessions. Central lines indicate the median value, and box edges represent the upper and lower quartiles. Multiple comparison tests were based on Duncan’s test at *p* < 0.05. Different lower case letters above bars indicate significant differences among haplotypes.

**Table 1 plants-12-02163-t001:** Descriptive statistics for salt tolerance in 173 rice germplasms.

Phenotype	Year	Mean ± SE	Range	CV	H_B_^2^
GP	2020	0.55 ± 0.024	0.00~1.0	57.18%	83.89%
	2021	0.56 ± 0.019	0.02~0.97	44.14%	
GR	2020	0.91 ± 0.47	0.10~1.00	15.27%	52.27%
	2021	0.93 ± 0.072	0.33~1.00	10.26%	
SL	2020	3.32 ± 0.11	0.69~8.22	43.39%	85.44%
	2021	4.00 ± 0.12	1.03~8.77	39.41%	
RL	2020	1.87 ± 0.16	0.00~8.63	80.50%	89.05%
	2021	4.08 ± 0.22	0.36~11.03	70.57%	
GPR	2020	0.41 ± 0.025	−0.04~1.00	78.41%	58.61%
	2021	0.39 ± 0.020	−0.30~0.98	67.72%	
GRR	2020	0.07 ± 0.01	−0.13~0.90	117.30%	68.35%
	2021	0.01 ± 0.01	−0.57~0.44	124.30%	
SLR	2020	0.61 ± 0.01	0.21~0.93	26.87%	66.47%
	2021	0.53 ± 0.01	0.07~0.83	29.52%	
RLR	2020	0.78 ± 0.02	−0.15~1.00	35.18%	80.51%
	2021	0.61 ± 0.02	−0.65~0.97	49.05%	
RSD	2020	0.24 ± 0.02	−0.03~0.95	81.19%	46.26%
	2021	0.21 ± 0.01	−0.14~0.73	67.60%	
CRS	2020	0.7 ± 0.02	0.07~0.96	29.32%	66.87%
	2021	0.56 ± 0.02	−0.20~0.88	37.29%	

GP: germination potential; GPR: germination potential relative to salt damage rate; GR: germination rate; GRR: germination rate relative to salt damage rate; SL: seedling length; SLR: seedling length relative salt damage rate; RL: root length; RLR: root length relative salt damage rate; RSD: relative salt damage rate at the germination stage; CRS: comprehensive relative salt damage rate in the early seedling stage.

**Table 2 plants-12-02163-t002:** Correlation analysis of salt tolerance among 173 rice germplasms.

Traits	GP	GPR	GR	GRR	SL	SLR	RL	RLR	RSD	CRS
GP	1									
GPR	−0.840 **	1								
GR	0.366 **	−0.309 **	1							
GRR	−0.261 *	0.321 **	−0.696 **	1						
SL	0.004	−0.023	0.201	−0.179	1					
SLR	−0.008	0.050	−0.240	0.219	−0.841 **	1				
RL	−0.166	0.180	0.105	−0.143	0.740 **	−0.722 **	1			
RLR	0.188	−0.182	−0.057	0.145	−0.648 **	0.662 **	−0.887 **	1		
RSD	−0.818 **	0.868 **	−0.570 **	0.538 **	−0.065	0.079	0.095	−0.105	1	
CRS	0.129	−0.110	−0.121	0.187	−0.777 **	0.836 **	−0.901 **	0.961 **	−0.043	1

Note: ** Significant correlation at the 0.01 level (double-tailed). * Significant correlation at the 0.05 level (double-tailed).

**Table 3 plants-12-02163-t003:** SNP loci associated with salt tolerance in 173 rice accessions in 2020 and 2021.

Trait	QTLs	Chr.	Position	*p* Value	R^2^ (%)	Year	Cloned Genes	Reference
GPR	*qGPR1-1*	1	38,365,940	1.07 × 10^−4^	11.3	2020		
	*qGPR1-2*	1	38,309,966	3.33 × 10^−5^	12.8	2020		
	*qGPR2-1*	2	24,709,239	7.28 × 10^−5^	11.8	2020		
			24,709,239	2.04 × 10^−4^	10.4	2021		
	*qGPR2-2*	2	35,432,245	8.33 × 10^−5^	11.6	2021	*OsMPS*	Schmidt et al. [33]
	*qGPR6*	6	7,793,840	1.58 × 10^−4^	10.8	2021		
	*qGPR7*	7	3,448,084	2.09 × 10^−4^	10.4	2021		
SLR	*qSLR2*	2	9,878,862	5.04 × 10^−5^	10.1	2020		
	*qSLR9*	9	17,238,271	4.58 × 10^−5^	10.2	2020		
			17,238,271	6.99 × 10^−5^	11.6	2021		
RSD	*qRSD1*	1	38,309,966	6.36 × 10^−5^	12.0	2020		
	*qRSD2*	2	4,515,468	6.03 × 10^−5^	11.7	2021		
	*qRSD3*	3	5,919,379	3.82 × 10^−5^	12.6	2020		
	*qRSD12*	12	23,873,291	5.53 × 10^−5^	11.9	2021		

GPR: germination potential relative to salt damage rate; SLR: seedling length relative to salt damage rate; RSD: relative salt damage rate at the germination stage.

**Table 4 plants-12-02163-t004:** Candidate genes and functional annotation.

QTL	Chr.	Gene ID	MSU ID	Feature Notes
	2	*Os02g0619600*	*LOC_Os02g40664*	zinc finger family protein, putative, expressed
	2	*Os02g0620100*	*LOC_Os02g40680*	mis12 protein, expressed
	2	*Os02g0621100*	*LOC_Os02g40770*	SET domain containing protein, expressed
*qGPR2*	2	*Os02g0621300*	*LOC_Os02g40784*	WAX2, putative, expressed
	2	*Os02g0621500*	*LOC_Os02g40810*	Zinc finger, ZZ type domain containing protein, expressed
	2	*Os02g0622100*	*LOC_Os02g40860*	CK1_CaseinKinase_1.5—CK1 includes the casein kinase 1 kinases, expressed
	2	*Os02g0622400*	*LOC_Os02g40890*	GLTP domain containing protein, putative, expressed
	2	*Os02g0622500*	*LOC_Os02g40900*	RNA recognition motif containing protein, putative, expressed
	9	*Os09g0456100*	*LOC_Os09g28300*	remorin C-terminal domain containing protein, putative, expressed
	9	*Os09g0456200*	*LOC_Os09g28310*	bZIP transcription factor, putative, expressed
	9	*Os09g0456700*	*LOC_Os09g28340*	expressed protein
*qSLR9*	9	*Os09g0456900*	*LOC_Os09g28370*	retrotransposon protein, putative, unclassified, expressed
	9	*Os09g0458000*	*LOC_Os09g28450*	paramyosin, putative, expressed
	9	*Os09g0458100*	*LOC_Os09g28460*	xyloglucan fucosyltransferase, putative, expressed
	9	*Os09g0458400*	*LOC_Os09g28480*	expressed protein

GPR: germination potential relative to salt damage rate; SLR: seedling length relative to salt damage rate.

## Data Availability

Not applicable.

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
