# Peer review of "Mining Salt Tolerance SNP Loci and Prediction of Candidate Genes in the Rice Bud Stage by Genome-Wide Association Analysis"

_plants, 2023, doi:10.3390/plants12112163_

Round 1
Reviewer 1 Report
I have some minor concerns before it can be accepted.
(1) P1-P2. “To date, approximately 439 QTLs related to ……”. It is better to add the reference or give a list of the 439 QTLs, 79 salt-tolerance genes and 13 salt-tolerance genes that the author mentioned.
(2) P2. “……in response to salt or ABA and GA treatment”. It is hard to understand how OsGASR1 expression were influenced by salt, ABA and GA.
(3) P2. “SKCL gene” should be “SKC1 gene”.
(4) P4. As shown in figure 2, the concentration of NaCl were marked as 0.5%, however, the description in text were 140 mmol/L NaCl. Besides, adding the data statistic of root or shoot length in figure 2 may make the results more convictive.
(5) It is better to add some references to elucidate the relationship between the ten index and salt stress resistance, especially RSD and CRS.
(6) Where is figure 3e?
(7) Through the correlation analysis of the two-year seedling stage, the authors concluded that GPR, SLR and CRS is appropriate for genome-wide associations. So, it is more reasonable to just show the association data of the three indicators in Table 3. The association data of other indicators can be listed in the Supplementary figures.
(8) If possible, the inducible or inhibitory of the three predicted genes under salt stress may help for verifying their biological functions.
(9) It is better to move Figure 9 and Figure 10 to supplementary figure because they were only mentioned in Materials and Methods.
Please carefully review the manuscript and pay attention to the correct written of genes and proteins, such as OSZB8 and OsNAC5. Besides, other written formats such as Arabidopsis, japonica and indica should be corrected throughout the manuscript.
Author Response
Response to the referees’ comments:
I have some minor concerns before it can be accepted.
(1) P1-P2. “To date, approximately 439 QTLs related to ……”. It is better to add the reference or give a list of the 439 QTLs, 79 salt-tolerance genes and 13 salt-tolerance genes that the author mentioned.
Answer: Thank you. We have added references to the corresponding parts of this revised manuscript.
(2) P2. “……in response to salt or ABA and GA treatment”. It is hard to understand how OsGASR1 expression were influenced by salt, ABA and GA.
Answer: Thanks for your advice. We have provided a more detailed description and explanation of this section in this revised version.
(3) P2. “SKCL gene” should be “SKC1 gene”.
Answer: Thanks for your advice. We have corrected this error in this revised manuscript.
(4) P4. As shown in figure 2, the concentration of NaCl were marked as 0.5%, however, the description in text were 140 mmol/L NaCl. Besides, adding the data statistic of root or shoot length in figure 2 may make the results more convictive.
Answer: Thanks for your comments. We have changed “0.5%” to “140 mmol/L NaCl” in this revised manuscript. The data statistic of root or shoot length has added as the Figure2b-2e in this revised manuscript.
(5) It is better to add some references to elucidate the relationship between the ten index and salt stress resistance, especially RSD and CRS.
Answer: Thanks for your comments. We have provided some references to elucidate the relationship between the ten index and salt stress resistance in this revised manuscript.
(6) Where is figure 3e?
Answer: We are sorry that there were two Figure 3c in the last manuscript. And we have corrected the errors in this revised manuscript.
(7) Through the correlation analysis of the two-year seedling stage, the authors concluded that GPR, SLR and CRS is appropriate for genome-wide associations. So, it is more reasonable to just show the association data of the three indicators in Table 3. The association data of other indicators can be listed in the Supplementary figures.
Answer: Thank you for your reviews. According to your comments and other studies, we have just showed the association data of the three indicators in Table 3 including GRP, SLR and RSD in this revised manuscript. The results of other indicators were listed as the supplementary Figure S1 and Table S1.
(8) If possible, the inducible or inhibitory of the three predicted genes under salt stress may help for verifying their biological functions.
Answer: Thank you. Yes, the inducible or inhibitory of the three predicted genes under salt stress is very important for verifying their biological functions. We will try our best to finish them in our future study. If we finished them, we hope it can be published as the continuous results of this study.
(9) It is better to move Figure 9 and Figure 10 to supplementary figure because they were only mentioned in Materials and Methods.
Answer: Thanks for your advice. We have moved figure 9 and figure 10 to supplementary Figure S2 and Figure S3, respectively.
Please carefully review the manuscript and pay attention to the correct written of genes and proteins, such as OSZB8 and OsNAC5. Besides, other written formats such as Arabidopsis, japonica and indica should be corrected throughout the manuscript.
Answer: This issue was overlooked by our negligence. Thank you for your reminder. We have made corresponding modifications to the correct writing of the genes, proteins and other written formats in this revised manuscript.
Reviewer 2 Report
Overall, the study appears to be well-designed and executed. However, a few areas could benefit from further clarification or improvement.
The abstract is well-written and provides a clear overview of the study. Here are some more detailed comments to further improve it.
You would add a more detailed explanation of why salt tolerance is important in rice breeding. It would also be helpful to briefly mention the current approaches to improving salt tolerance in rice and how this study fits into that context.
In the methods section, it would be useful to provide more information on the criteria used to select the 173 rice accessions for the study. Additionally, it would be helpful to mention how the SNP data was obtained.
The results section could be improved by providing more specific details on the QTLs that were identified. For example, what are the chromosomal locations of these QTLs, and what is the magnitude of their effect on salt tolerance traits?
Introduction
The second sentence provides important context by noting that rice is a nonhalophytic, salt-intolerant plant. However, the sentence could be improved by briefly explaining what is meant by "nonhalophytic" and why it is relevant to the discussion of salt stress.
It would be helpful to briefly explain how salt tolerance genes can be used to cultivate salt-tolerant rice varieties.
Page 2: The first category of genes: It would be helpful to provide more specific details on the mechanisms of action of these genes and any known interactions with other genes or pathways involved in salt tolerance.
The second category of genes: The authors may want to add more details on how these genes regulate salt tolerance. For example, explaining how SNAC2 encodes plant-specific NAC transcription factors and how these factors enhance salt tolerance would be useful.
The third type of genes: For each gene mentioned, it would be nice to briefly explain the mechanism by which it regulates ion transport and salt tolerance.
Results
Figures: low quality and resolution
Discussion
Please avoid repetition of the results and focus more on discussing the findings in detail. It would be helpful to compare the results with previous studies to provide context and highlight the novelty of the findings. Additionally, it would be useful to discuss the potential implications of the findings and their significance for future research and practical applications. It is worth noting that this study only focused on salt tolerance during the germination stage, and future studies should explore salt tolerance at different growth stages to fully understand the genetic mechanisms underlying salt tolerance in rice. Additionally, the contribution of environmental factors to salt tolerance-related traits was not considered in this study, and it would be interesting to investigate the interaction between genetic and environmental factors in future studies.
Methods
Please provide more details about the methods used to measure the salt concentration and how it was applied to the seeds in the treatment group. Why was a salt concentration of 140 mmol/L chosen for the salt stress control group?
Genotype Identification: Were any quality control measures taken during the sequencing process, and how was the resulting data analysed?
Structure Analysis: I would like to know the parameters used for the PLINK software and the STRUCTURE analysis, as well as the online tools used to analyse and visualise the results.
Genome-Wide Association Analysis: I’d see more information on the phenotype(s) studied and the rationale behind the choice of GLM and MLM. Additionally, it would be useful to briefly explain the Bonferroni and Benjamini-Hochberg correction methods for readers who may not be familiar with these statistical techniques.
Minor editing of English language required
Author Response
Response to the referees’ comments:
Overall, the study appears to be well-designed and executed. However, a few areas could benefit from further clarification or improvement.
The abstract is well-written and provides a clear overview of the study. Here are some more detailed comments to further improve it.
You would add a more detailed explanation of why salt tolerance is important in rice breeding. It would also be helpful to briefly mention the current approaches to improving salt tolerance in rice and how this study fits into that context.
Answer: First of all, thank you for your recognition of our research. We have provided a more detailed description and correction of abstract section based on your comments in this revised manuscript.
In the methods section, it would be useful to provide more information on the criteria used to select the 173 rice accessions for the study. Additionally, it would be helpful to mention how the SNP data was obtained.
Answer: Thank you. We have explained the reasons for selecting 173 rice materials and provided information to the acquisition of SNP data in this revised manuscript.
The results section could be improved by providing more specific details on the QTLs that were identified. For example, what are the chromosomal locations of these QTLs, and what is the magnitude of their effect on salt tolerance traits?
Answer: We have improved the relevant details of the identified QTL, including more detailed information such as chromosome position, their effect on salt tolerance traits and so on. This will help readers better understand our study.
Introduction
The second sentence provides important context by noting that rice is a nonhalophytic, salt-intolerant plant. However, the sentence could be improved by briefly explaining what is meant by “nonhalophytic” and why it is relevant to the discussion of salt stress.
Answer: We have added relevant references and provided further explanations of “nonhalophytic” in this revised manuscript.
It would be helpful to briefly explain how salt tolerance genes can be used to cultivate salt-tolerant rice varieties.
Answer: We have briefly explained how salt tolerance genes can be used to cultivate salt-tolerant rice varieties in this revised manuscript.
Page 2: The first category of genes: It would be helpful to provide more specific details on the mechanisms of action of these genes and any known interactions with other genes or pathways involved in salt tolerance.
The second category of genes: The authors may want to add more details on how these genes regulate salt tolerance. For example, explaining how SNAC2 encodes plant-specific NAC transcription factors and how these factors enhance salt tolerance would be useful.
The third type of genes: For each gene mentioned, it would be nice to briefly explain the mechanism by which it regulates ion transport and salt tolerance.
Answer: Thank you. We have provided more detailed descriptions of these three types of genes in the introduction, including the pathways and mechanisms involved in salt tolerance in this revised manuscript.
Results
Figures: low quality and resolution
Answer: We have improved the quality and resolution of the low quality figures including Figure 1, Figure2, Figure 3, Figure 4 and Figure 5.
Discussion
Please avoid repetition of the results and focus more on discussing the findings in detail. It would be helpful to compare the results with previous studies to provide context and highlight the novelty of the findings. Additionally, it would be useful to discuss the potential implications of the findings and their significance for future research and practical applications. It is worth noting that this study only focused on salt tolerance during the germination stage, and future studies should explore salt tolerance at different growth stages to fully understand the genetic mechanisms underlying salt tolerance in rice. Additionally, the contribution of environmental factors to salt tolerance-related traits was not considered in this study, and it would be interesting to investigate the interaction between genetic and environmental factors in future studies.
Answer: Thank you. We have further explored and supplemented other research findings regarding the issue you raised. Finally, we provided prospects for future research including salt tolerance at different growth stages and the contribution of environmental factors to fully understand the genetic mechanisms underlying salt tolerance in rice.
Methods
Please provide more details about the methods used to measure the salt concentration and how it was applied to the seeds in the treatment group. Why was a salt concentration of 140 mmol/L chosen for the salt stress control group?
Answer: Thank you. We have explained in the method section of the article why 140mmol/L was selected as the salt stress control group. We also have further described the processing details of measuring salt concentration in this revised manuscript.
Genotype Identification: Were any quality control measures taken during the sequencing process, and how was the resulting data analysed?
Answer: Thank you. We have provided detail information about quality control measures and resulting data analysis in this revised manuscript. To ensure data quality, it is necessary to filter the original data before information analysis to reduce the analysis interference caused by invalid data. First, we use fastp for quality control of offline raw reads, and filter low-quality data to get clean reads.
The steps for filtering reads are as follows: 1) Remove reads containing adapter 2) Remove reads with N content greater than 10% 3) Remove reads with all A base 4) Remove low-quality reads (base numbers with a quality value of Q ≤ 20 account for more than 50% of the entire read).
Then, the gVCF file was generated after SNP Calling. Multiple sample gVCF files are merged to generate a VCF file, which was used as the genotype data in this study.
Structure Analysis: I would like to know the parameters used for the PLINK software and the STRUCTURE analysis, as well as the online tools used to analyse and visualise the results.
Answer: Thank you. We have provided the parameters used for the PLINK software and the STRUCTURE analysis in Table S4 and Table S5, respectively. We also have provided the information of online tools used to analyze and visualise the results in this revised manuscript.
Genome-Wide Association Analysis: I’d see more information on the phenotype(s) studied and the rationale behind the choice of GLM and MLM. Additionally, it would be useful to briefly explain the Bonferroni and Benjamini-Hochberg correction methods for readers who may not be familiar with these statistical techniques.
Answer: Thank you. We have briefly explained the Bonferroni and Benjamini-Hochberg correction methods in this revised manuscript.
Minor editing of English language required
Answer: We have improved the English language by professional English editing service.